# Combining 675 nm Laser with Isotretinoin for Enhanced Acne Vulgaris Treatment Outcomes

**DOI:** 10.3390/healthcare13233068

**Published:** 2025-11-26

**Authors:** Ariel Haus, Alessandro Clementi, Giovanni Cannarozzo, Luca Guarino, Elena Zappia, Marco Gratteri, Annunziata Dattola, Steven Paul Nisticò

**Affiliations:** 1Department of Dermatology, Haus Dermatology, London W1G 8QL, UK; drhaus@drhausdermatology.com; 2Department of Dermatology, University of Modena and Reggio Emilia, 41124 Modena, Italy; 3Department of Clinical Sciences, Sapienza University of Rome, 00185 Rome, Italy; drcannarozzo@gmail.com (G.C.); luca.guarino@uniroma1.it (L.G.); annunziata.dattola@uniroma1.it (A.D.); steven.nistico@uniroma1.it (S.P.N.); 4Department of Plastic, Reconstructive and Cosmetic Surgery, Campus Bio-Medico University Hospital, 00185 Rome, Italy; m.gratteri@unicampus.it

**Keywords:** acne vulgaris, 675 nm laser, isotretinoin, combination treatment

## Abstract

**Background:** Acne vulgaris is a multifactorial disease with significant clinical and psychosocial impacts. **Methods:** The purpose of this study was to evaluate the efficacy of a combination therapy including oral isotretinoin and a non-ablative 675 nm red-light laser compared with laser monotherapy. Thirteen young patients with active facial acne of varying severity were enrolled and divided into two groups: Seven subjects received laser monotherapy treatment (RT group), while six underwent combined laser and isotretinoin therapy (RTI group). The laser protocol consisted of six weekly sessions, with a 5-point pain scale used to monitor tolerability. Standardized photographs were obtained at baseline and at a 3-month follow-up after the last treatment. Each patient completed the Acne Radar Questionnaire, and lesion severity was assessed using the Global Evaluation Acne (GEA) scale. **Results:** All patients completed the study without adverse events. Scores from the Acne Radar Questionnaire improved in both groups, while the GEA scale demonstrated a significant reduction in lesion severity, confirmed by photographic comparison. Pain was reported as mild in most cases, and no discontinuations occurred. **Conclusions:** These findings indicate that the 675 nm laser is a safe and effective therapeutic option for acne vulgaris, with isotretinoin addition resulting in more rapid and pronounced clinical improvement.

## 1. Introduction

A common inflammatory skin disorder that affects the pilosebaceous unit and frequently lasts over time is acne vulgaris, also known as acne. Acne vulgaris affects a significant percentage of the population, with up to 85% of young people experiencing this condition. The typical age range for acne onset is between 14 and 19 years in males and 10 to 17 years in females [1]. Although it is most seen in adolescents, people of various ages (in pre-adolescents and even in post-adolescents) can be impacted [2,3]. Acne vulgaris is a multifactorial skin condition whose etiopathogenesis involves hormonal, inflammatory, and microbial factors. The increase in androgen hormones during puberty stimulates the sebaceous glands to produce excess sebum. The combination of seborrhoea and follicular hyperkeratinisation causes obstruction of the hair follicles with the formation of comedones. The resulting anaerobic environment promotes the proliferation of Cutibacterium acnes, which induces a local inflammatory response and the subsequent appearance of papules and pustules [4,5]. The lesions mainly affect the face but can extend to the trunk and upper limbs. The severity varies from mild forms with few comedones to severe cases with scarring, hyperpigmentation, and significant psychological impact [4,5].

Although acne vulgaris affects individuals of all ethnicities, its incidence and severity can vary based on genetic, environmental, and behavioural factors; a lower prevalence has been reported among individuals of Asian and African origin [6]. The disease is commonly classified as mild, moderate, or severe, depending on the number and type of lesions. Mild forms present with non-inflammatory comedones, while moderate and severe forms include papules, pustules, and inflammatory nodules, often followed by scarring: atrophic, which is more common, or hypertrophic and keloid, due to excessive collagen production. The most affected areas are the face, back, and chest, which have a high density of sebaceous glands [7].

Acne has a significant impact on quality of life, affecting self-esteem, social relationships, and psychological well-being. Prompt treatment is, therefore, recommended, especially in inflammatory forms [8]. Combined topical and systemic therapies are the standard for moderate to severe forms. Systemic treatments include antibiotics, hormonal agents, and isotretinoin [9]. However, each option has specific limitations: antibiotics can cause gastrointestinal disorders and bacterial resistance, hormonal agents increase the risk of arrhythmias and thrombosis, while isotretinoin, although effective in 60% of cases, requires rigorous monitoring due to its potential teratogenicity and psychological side effects [10,11].

The use of lasers for the management of acne scars is now well established in the literature, with multiple wavelengths demonstrating efficacy in remodelling dermal architecture and improving long-term outcomes [12,13,14]. Given these results, their application has also been explored in active acne, where device-based treatments may act as alternative or complementary options to traditional pharmacologic therapies. These treatments often target specific mechanisms of acne, such as bacterial growth, inflammation, and excess sebum production.

Pulsed dye laser (PDL) targets the blood vessels in the skin to reduce inflammation and redness, which can help with acne-related swelling and scarring [15]. Blue light therapy (photodynamic therapy) specifically targets the bacteria responsible for acne (*Cutibacterium acnes*), reducing bacterial load and inflammation [16]. Recent research has shown that both the 1064 nm Nd:YAG laser and intense pulsed light (IPL) offer outstanding cosmetic and therapeutic benefits. In particular, IPL uses broad-spectrum light to target the skin’s deeper layers, reducing oil production and treating acne lesions [17,18].

Red light with a wavelength of 675–685 nm is utilized in laser treatments for several skin disorders, including acne vulgaris [19]. This specific wavelength has been shown to penetrate the skin effectively, promoting healing by stimulating cellular processes such as collagen production and reducing inflammation, which is a significant factor in acne development [20,21,22]. The treated skin improves in smoothness, softness, “glow effect”, and texture, with minimal recovery time, also in individuals with darker skin phenotypes [23,24].

Additionally, preclinical research has demonstrated the efficacy of the 675 nm non-ablative laser. Specifically, histological results from human skin biopsies and cultured human fibroblasts have demonstrated that the laser stimulates collagen remodelling, leading to a notable increase in thin and new fibres [25,26].

Device-based treatments can precisely target specific skin areas, which are particularly useful for localized acne or scarring. For many patients, these modalities represent an effective adjunct or alternative to traditional acne medications, especially when looking to reduce acne. The results of device-based treatments can vary from person to person, and they may not be as effective as oral medications or topical therapies.

The intricate combination of the several pathogenic elements that define acne vulgaris complicates its treatment, and a more thorough and frequently quicker improvement is achieved with combination therapies [27].

Isotretinoin acts in a dose- and time-dependent manner by reducing sebocyte activity, normalising keratinisation, and modulating inflammatory cytokines [9]. The 675 nm laser, through selective photothermolysis, influences fibroblasts, keratinocytes, and macrophages, promoting dermal remodelling and reducing inflammation [19].

Given this complementary biological action, the present study evaluated the clinical efficacy of the combination of isotretinoin and non-ablative 675 nm laser compared to laser monotherapy in patients with active acne vulgaris.

## 2. Materials and Methods

The study population consisted of 13 young patients with a mean age of 24.5 ± 7.3 years old and a diagnosis of active acne in the face of varying severity. The patients were divided into two groups: the first group, consisting of 7 subjects, underwent laser-only treatment (RT group); the second group, consisting of 6 patients, underwent combined laser and isotretinoin treatment (RTI group). The characteristics of the two groups in terms of age, sex, and skin type are described in Table 1. Given the exploratory nature of this preliminary clinical evaluation, patients were divided according to treatment modality rather than demographic or hormonal parameters.

Patients in both groups met the inclusion criteria (presence of active acne in the face), while the exclusion criteria included incomplete treatment and follow-up less than one month after the end of treatment. Patients with previous systemic isotretinoin or laser therapy within six months prior to enrolment were excluded.

All patients in both groups were treated with a 675 nm fractional laser (RedTouch PRO; Deka M.E.L.A, Calenzano, Italy) every 7 days, for a total of 6 sessions. The laser emits light at a wavelength of 675 nm, enabling the absorption of energy by dermal collagen and, to a lesser extent, by melanin. The RedTouch PRO laser system is equipped with a 13 × 13 mm scanning system, integrated in the handpiece, able to generate microzones (DOT) of either sub-ablative or selective thermal damage on the skin, preserving the epidermal layer thanks to a 5 °C integrated skin cooling. The scanner can be used in two modes: standard and moveo mode [28]. With the standard mode, the handpiece must be positioned perpendicular to the skin surface and must stay in the same position until the system has finished the entire scanning area. After that, the handpiece must be moved in such a way that there is no overlap between several scanning areas. The moveo mode, on the other hand, consists of using the handpiece in motion, performing a single line of microzones of thermal damage. In this study, the standard mode was selected and evaluated. The RTI group had combined laser and isotretinoin treatment. In the RTI group, isotretinoin was administered orally at 0.1–0.3 mg/kg/day, starting concurrently with laser therapy. The laser parameters were set according to the patient’s skin type to avoid or mitigate possible adverse effects.

Specifically, Table 2 shows the ranges of power, DOT pulse duration (dwell time), spacing between each DOT (spacing), number of pulses delivered consecutively over the same point (SmartStack), and energy per DOT (energy/DOT) used according to skin type; as the Fitzpatrick grade increases, lower powers and dwell time are used, with spacing tending to be higher. All possible adverse effects were monitored during the study.

The patient’s perception of pain during treatment was assessed using a 5-point scale (0: no pain, 1: mild pain, 2: moderate pain, 3: severe pain, 4: very severe pain).

Clinical images were acquired before treatment and at a follow-up of 3 months after the last treatment session.

The Acne Radar Questionnaire was administered to each patient before treatment and at a 3-month follow-up after the last treatment session. This questionnaire contains 10 items that cover three areas: objective symptoms (such as imperfection, itchiness, and insomnia), subjective symptoms (such as depression, discomfort, lack of happiness, and self-confidence), and relational difficulties (such as social/working relations and intimate relations). For each question, patients are asked to rate how significantly acne interferes with their everyday activities using a scale of 1 to 10 (1: strongly disagree, 10: completely agree). All these data were represented by using a radar graph.

The radar graph is a graphical representation of data from numerous variables. It consists of a series of rays that originate in the centre and form equal angles between themselves; each ray is related to a variable (item). The patient profile is obtained by connecting all locations related to each topic on each ray with straight lines [29].

The parameters of the questionnaire were divided into quartiles. The first, second, and third quartiles were calculated and graphed for each parameter in each group, both before treatment and after a 3-month follow-up after the last treatment session.

The Global Acne Severity (GEA) scale, a tool that has been proven to be effective in measuring acne severity, was utilised to assess the progression of the acne lesions. The GEA scale employs a global evaluation of acne severity, categorised as follows: grade 0, clear and no lesions; grade 1, almost clear and almost no lesions; grade 2, less than half of the face is affected; grade 3, more than half of the face is involved; grade 4, entire face is affected; and grade 5, highly inflammatory and nodules [30]. This acne assessment scale was completed by the physician before treatment and at a 3-month follow-up after the last treatment session.

The Student’s *t*-test was used to statistical analysis of clinical data. The level of significance for statistical tests was 0.05.

## 3. Results

All patients completed the treatment without reporting any adverse effects. Only a proportion of subjects in both groups experienced mild pain during treatment (57% no pain and 43% mild pain for the RT group, 67% no pain, and 33% mild pain for the RTI group).

Mean scores for all Acne Radar Questionnaire items improved in both groups. For the RT group, there were significant decreases in both objective parameters (imperfections from 8.2 ± 1.7 pre-treatment to 5.4 ± 2.4 post-treatment, *p*-value < 0.05) and subjective parameters (self-confidence from 8.0 ± 2.9 pre-treatment to 4.5 ± 2.1 post treatment, *p*-value < 0.05; happiness from 6.6 ± 3.1 pre-treatment to 4.3 ± 3.0 post-treatment, *p*-value < 0.05; discomfort from 5.6 ± 3.0 pre-treatment to 3.0 ± 2.3 post-treatment, *p*-value < 0.05).

The RTI group also recorded significant improvements in objective parameters (imperfections from 8.0 ± 1.3 pre-treatment to 3.7 ± 3.7 post-treatment, *p*-value < 0.05; itchiness from 5.0 ± 1.3 pre-treatment to 2.8 ± 2.5 post-treatment, *p*-value < 0.05; insomnia 3.3 ± 2.2 pre-treatment to 1.5 ± 0.8 post-treatment, *p*-value < 0.05), subjective parameters (happiness from 6.8 ± 2.6 pre-treatment to 2.5 ± 1.8 post-treatment, *p*-value <0.05), and relational difficulties (work and study relations from 5.0 ± 2.9 to 2.0 ± 1.2, *p*-value < 0.05).

In addition, the analyses obtained from the quartiles showed improvements in most variables for the RT group, and all variables for the RTI group, as shown in Figure 1, Figure 2 and Figure 3.

Regarding the GEA scale, significant decreases were recorded for both groups, indicating improvement of acne lesions in all treated subjects. For the RT group, the mean GEA scale scores significantly (*p* < 0.05) decreased from 1.9 ± 0.9 at baseline to 1.4 ± 0.9 at 3 MFU after the last treatment session, while for the RTI group, the mean GEA scale scores significantly (*p* < 0.05) decreased from 2.5 ± 1.0 at baseline to 0.8 ± 0.9 at 3 MFU after the last treatment session (Figure 4).

Finally, photographic images confirmed the positive results obtained with both treatments, as shown in Figure 5 and Figure 6 for the RT group, and Figure 7 for the RTI group. The heterogeneity of the sample, including differences in sex and age distribution, was acknowledged as a potential confounding factor when interpreting the results.

## 4. Discussion

Acne vulgaris is a chronic inflammatory condition with multiple causes, in which hormonal, microbial, and inflammatory factors interact to cause inflammation, scarring, and psychological impact, especially in adolescents. Isotretinoin, introduced about forty years ago, is the only oral therapy capable of acting on all the pathogenic mechanisms of acne, reducing sebum production, the proliferation of Cutibacterium acnes, and follicular keratinisation, as well as exerting a direct anti-inflammatory effect [31,32].

Its main mechanism of action consists of inhibiting sebaceous secretion through isomerisation to all-trans retinoic acid (ATRA), which binds to retinoid receptors and induces the expression of FoxO and TRAIL, causing apoptosis of sebocytes. In addition, isotretinoin modulates the immune response by increasing the production of anti-inflammatory cytokines and reducing that of MMP-8 and MMP-9 metalloproteinases, markers of inflammation [33,34,35,36,37,38,39].

This study aimed to assess, within the methodological limits of this exploratory setting, the clinical efficacy and safety profile of a 675 nm diode laser in the treatment of moderate to severe acne vulgaris, both as a monotherapy and in combination with systemic isotretinoin. Our findings show that, although laser monotherapy produced a statistically and clinically significant improvement in acne lesion severity, the combined therapeutic method (laser plus isotretinoin) can achieve even greater clinical results. The enhanced outcomes observed in the combination group may be attributed to the complementary mechanisms and synergistic effects of action of the 675 nm laser and isotretinoin.

The 675 nm wavelength selectively targets dermal chromophores, particularly collagen and melanin, with minimal epidermal disruption [19]. It facilitates dermal remodelling, reduces perifollicular inflammation, and modulates sebaceous gland activity through selective photothermolysis [21]. This mechanism, when paired with isotretinoin, a retinoid that acts on multiple pathogenic factors of acne including sebaceous gland atrophy, normalization of follicular keratinization, and reduction in Cutibacterium acnes proliferation, creates a synergistic therapeutic effect [40]. Together, these mechanisms likely enhance the overall therapeutic response and accelerate clinical improvement [41].

Published research and clinical experience clearly show that oral isotretinoin therapy is the most often used treatment for moderate-to-severe acne vulgaris. However, patients should be informed that it has several typical adverse effects, some of which can be less severe and occur less frequently with appropriate usage [42].

Nevertheless, it is important to emphasize the significant efficacy demonstrated by the 675 nm laser as a standalone treatment. Both the laser-only and the combination groups showed a marked reduction in lesions, accompanied by improved skin texture and decreased erythema. These findings support the laser’s utility as a valuable monotherapy, particularly in patients who are not candidates for systemic treatment due to contraindications, intolerance, or personal preference [42].

Furthermore, the safety profile of the 675 nm laser, both alone and in combination with isotretinoin, was favourable. No severe adverse effects or long-term complications were observed, reaffirming the device’s suitability for use in routine clinical settings, including in conjunction with systemic retinoids; a practice that has been previously approached with caution [19,43]. Importantly, the tolerability of the combined treatment was acceptable, with no unexpected adverse effects reported, suggesting that certain non-ablative laser modalities can be safely employed concurrently with isotretinoin when appropriate precautions are taken, challenging historical contraindications [43].

The 675 nm laser may help mitigate acne-related changes by reducing sebaceous gland size, regulating sebum production, and limiting the proliferation of acne-causing bacteria, ultimately improving the appearance and the texture of skin [41]. This latter effect is believed to be mediated through its ability to stimulate neocollagenesis and promote dermal remodelling. Histologically, exposure to this wavelength has been associated with increased collagen fibre density and organization, as well as a reduction in the expression of pro-inflammatory cytokines [25,26,44]. These mechanisms contribute to improved skin texture, enhanced pigmentation uniformity, and a decrease in the long-term sequelae of inflammatory acne lesions.

Red light acts through photobiochemical reactions that include increased production of adenosine triphosphate (ATP) (through absorption of mitochondrial protopor-phyrin IX), modulation of intracellular oxidative stress, induction of transcription factors, changes in collagen synthesis due to fibroblast growth factor (FGF) activation, increased type 1 procollagen, increased matrix metalloproteinase (MMP)-9, decreased MMP-1, stimulation of angiogenesis, increased blood flow, and a significant anti-inflammatory effect [45]. Specifically, this wavelength has been shown to downregulate the expression of key pro-inflammatory cytokines, including interleukin-1 beta (IL-1β), interleukin-6 (IL-6), and tumour necrosis factor-alpha (TNF-α), within the treated skin [21]. The suppression of these cytokines likely contributes to the observed reduction in perifollicular inflammation and may also play a role in preventing the progression to post-inflammatory erythema and scarring.

This anti-inflammatory effect, when combined with the laser’s ability to stimulate neocollagenesis and improve skin texture, positions the 675 nm laser as a promising therapeutic option not only for active acne lesions but also for the management of acne-related dyschromia and early atrophic scarring.

In this investigation, established instruments for data collecting and clinical evaluation were employed, including the GEA scale, which is renowned for its high inter-rater reliability [30]. Radar graphs allowed for a clear, comparative display of symptoms across time, among individuals, or between groups (e.g., male vs. female) [46]. Additionally, a photographic inquiry is an effective way to assess inflammatory acne [47].

The study found positive outcomes for both image and clinical scoring; indeed, our research demonstrates that moderate-to-severe acne may be significantly and sustainably reduced in a wide variety of skin types. Subjects showed an overall improvement in acne lesions appearance according to the GEA scale and Acne Radar Questionnaire findings, during the six well-tolerated 675 nm laser treatments and three months after the last laser treatment, suggesting that the sebaceous gland impact provided gradual and long-lasting improvement. These results of sustained improvement over time without additional therapy point to a mechanistic, sebaceous gland-modulating action influencing the disease’s natural course.

## 5. Limitations

Despite the encouraging results, this study’s limitations include a relatively small sample size and a brief follow-up period. The small sample size results in lower statistical power and increases the risk of type II error, as it may conceal real differences between groups that do not reach significance. The analysis was conducted for exploratory purposes, and the results should, therefore, be interpreted with caution, given the limited number of subjects.

To evaluate the durability of outcomes and the rate of recurrence after therapy, long-term research is required. It was not possible to perform subgroup analyses based on Fitzpatrick skin type, initial acne severity, or hormonal profile, which could influence treatment response. Furthermore, the short observation period does not allow for an estimation of the persistence of results over time or the recurrence rate.

Future research should consider the possibility that individual variances in Fitzpatrick skin type, hormonal profile, and acne severity, as well as individual variability in response to isotretinoin and laser therapy, might affect the results.

Prospective multicentre studies with larger samples, prolonged follow-up, and a control arm with isotretinoin monotherapy will be necessary to confirm the durability of the results and better clarify the specific contribution of each treatment, as well as the possible synergistic effect of the combination therapy.

## 6. Conclusions

The 675 nm laser is confirmed as a promising and well-tolerated treatment for acne vulgaris, capable of improving active lesions and skin texture quality. Combination with isotretinoin appears to accelerate the clinical response, suggesting a possible synergistic effect. Within the limitations of this exploratory study, these results indicate that the 675 nm laser, alone or in combination, may represent a useful therapeutic option, encouraging further investigation in prospective controlled studies.

## Figures and Tables

**Figure 1 healthcare-13-03068-f001:**
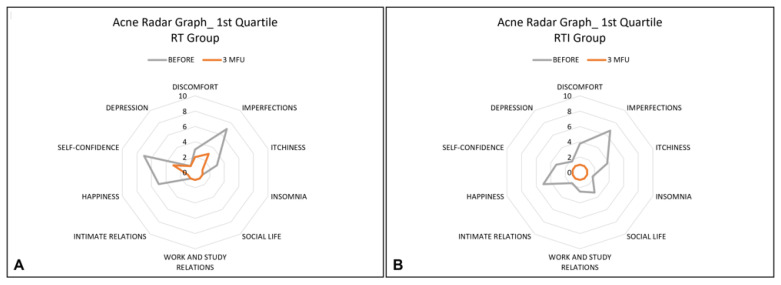
Mean distribution of the first quartile of all variables before (grey line) and post-treatment sessions (orange line) for the RT group (**A**) and for the RTI group (**B**). A reduction in perception of all variables for all patients of the RTI group was observed after the treatment.

**Figure 2 healthcare-13-03068-f002:**
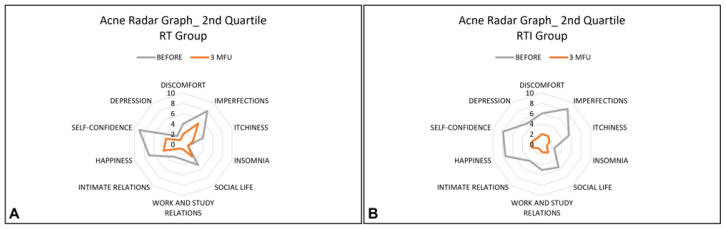
Mean distribution of the second quartile of all variables before (grey line) and post-treatment sessions (orange line) for the RT group (**A**) and for the RTI group (**B**). A reduction in perception of all variables for all patients of both groups was observed after the treatment.

**Figure 3 healthcare-13-03068-f003:**
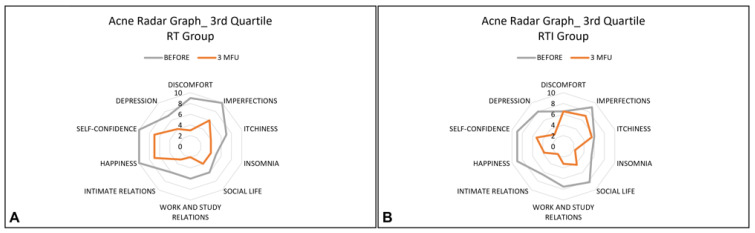
Mean distribution of the third quartile of all variables before (grey line) and post-treatment sessions (orange line) for the RT group (**A**) and for the RTI group (**B**). A reduction in perception of all variables for all patients of both groups was observed after the treatment.

**Figure 4 healthcare-13-03068-f004:**
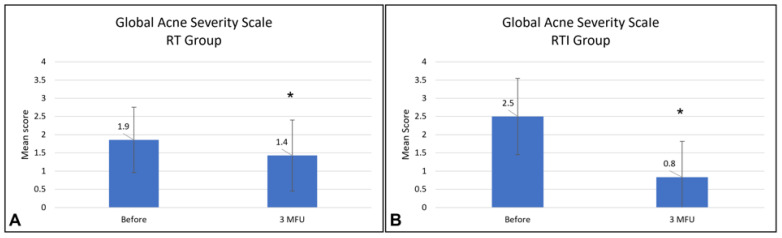
Mean GEA scores before and at 3 MFU after the last treatment session for the RT group (**A**) and for the RTI group (**B**). GEA, Global Acne Severity. Asterisks (*) indicate statistically significant differences compared with baseline (*p* < 0.05).

**Figure 5 healthcare-13-03068-f005:**
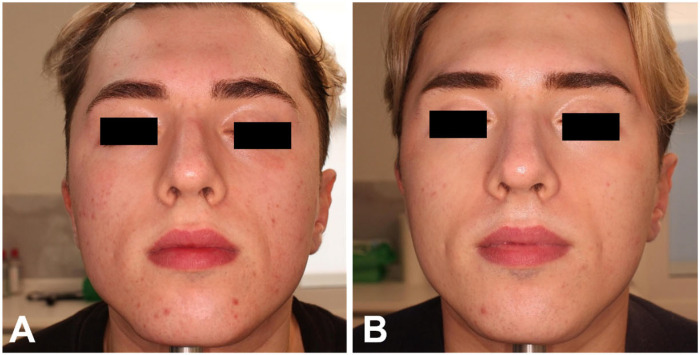
Frontal view of a male patient of the RT group before (**A**) and at 3 MFU after the last treatment session (**B**). Left profile of the same male patient before (**C**) and at 3 MFU after the last treatment session (**D**). Right profile of the same male patient before (**E**) and at 3 MFU after the last treatment session (**F**).

**Figure 6 healthcare-13-03068-f006:**
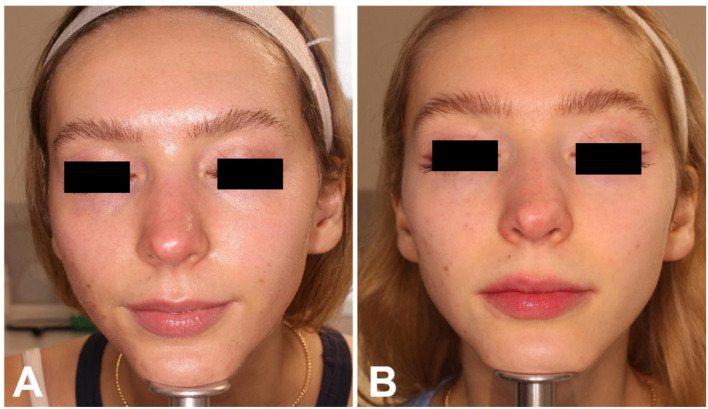
Frontal view of a female patient of the RT group before (**A**) and at 3 MFU after the last treatment session (**B**). Left profile of the same female patient before (**C**) and at 3 MFU after the last treatment session (**D**). Right profile of the same female patient before (**E**) and at 3 MFU after the last treatment session (**F**).

**Figure 7 healthcare-13-03068-f007:**
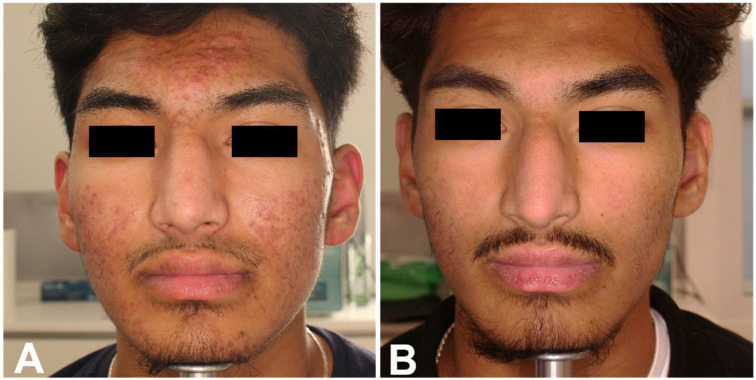
Frontal view of a male patient of the RTI group before (**A**) and at 3 MFU after the last treatment session (**B**). Left profile of the same male patient before (**C**) and at 3 MFU after the last treatment session (**D**). Right profile of the same male patient before (**E**) and at 3 MFU after the last treatment session (**F**).

**Table 1 healthcare-13-03068-t001:** Patient characteristics in terms of sex, age, and skin type of the two groups, RT and RTI.

Group	Sex	Age	Fitzpatrick Skin Type
RT group	86% females 14% males	28 ± 8.3 years	29% type II 42% type III 29% type IV
RTI group	33% females 67% males	20.3 ± 2.9 years	17% type I 66% type II 17% type V

**Table 2 healthcare-13-03068-t002:** Treatment parameters according to patient phototype.

Fitzpatrick Skin Type	Power (W)	Dwell Time (ms)	Spacing (μm)	SmartStack	Energy/DOT (J)	Number of Treatments	Time Interval Between Treatments
I	10	150	1500	1	1.5	6	7 Days
II	10	150	1500	1	1.5	6	7 Days
III	5–10	125–150	1500–2000	1	0.625–1.5	6	7 Days
IV	5–8	100–150	2000–2500	1	0.5–1.2	6	7 Days
V	5	125	1500	1	0.625	6	7 Days

## Data Availability

The data presented in this study are available on request from the corresponding author due to patient privacy and ethical restrictions.

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
