# Peer review of "Combining 675 nm Laser with Isotretinoin for Enhanced Acne Vulgaris Treatment Outcomes"

_healthcare, 2025, doi:10.3390/healthcare13233068_

Round 1
Reviewer 1 Report
Comments and Suggestions for Authors
The article presents an innovative approach to acne vulgaris by combining a 675 nm non-ablative laser with oral isotretinoin, showing synergistic and rapid improvement in clinical and psychosocial outcomes. The methods and results are generally well explained, with effective use of tables and graphics. However, the sample size is small and a longer follow-up would be needed to assess recurrence and durability of results.
The statistical analysis in the manuscript, while appropriately employing descriptive measures and inferential tests such as the Student’s t-test, is fundamentally limited by the small sample size (n=13 divided into two groups). A limited sample reduces statistical power, heightening the risk of type II error, in which true differences between groups may not reach statistical significance despite their presence. This limitation also precludes thorough subgroup analysis—such as by Fitzpatrick skin type, baseline acne severity, or hormonal status—which could reveal relevant treatment interactions or differential effects.​
The manuscript primarily presents mean changes and pre/post comparisons (e.g., using t-tests for the Acne Radar Questionnaire and GEA score). For small samples, the reliability and robustness of t-tests can be questionable, particularly if data distribution assumptions (normality, equal variances) are not met. With skewed data or outliers, results can over- or understate effect magnitude. Additionally, the study lacks reporting of effect sizes or confidence intervals, which makes clinical interpretation of statistical findings less precise and limits assessment of the likely variability if the study were repeated in larger cohorts.​
With such a limited number of participants, p-values alone are not wholly informative; small changes in individual outcomes can disproportionately affect group results, potentially exaggerating or masking true effects. For small n, the reporting of individual response profiles or nonparametric analyses could provide more nuanced insight.
Furthermore, the lack of correction for multiple comparisons across numerous questionnaire domains could inflate the risk of false-positive findings. For transparent reporting and more reliable evidence, larger multisite or longer-term studies, with pre-specified statistical plans including subgroup analysis and effect size reporting, would be beneficial.
Further details regarding patient selection and baseline characteristics are recommended. English language could be refined for consistency and fluency. Future research should address efficacy in broader, more diverse patient populations.
Author Response
We thank the reviewer for the methodological and statistical comments.
We have included the Limitations section to explicitly highlight the small sample size, the risk of type II error, and the exploratory nature of the analysis. The absence of subgroup analyses have been added, as well as a statement inviting caution in interpreting the results.
We have also emphasised the need for multicentre studies with larger samples, prolonged follow-ups, to confirm the durability of the results.
The language has been revised for greater clarity and readability throughout the article.
Reviewer 2 Report
Comments and Suggestions for Authors
Adequate sampling is very important for clinical practice.
The introduction contains a multitude of general information about the clinical and etiopathogenic aspects of acne. To more clearly identify the objectives of this research, it is necessary to specify the interaction of isotretinoin (dose-dependent, time-dependent) with the etiopathogenic mechanisms of acne. At the same time, it is necessary to clarify the mechanism of action of laser radiation on sebocytes, keratinocytes, macrophages, fibroblasts (cellular elements involved in the etiopathogenesis of acne). This allows us to understand the penetrability and effectiveness of laser radiation.
The Materials and Methods section presents the protocol from the technical sheet of the laser device. The small number of patients included in the study does not allow for an assessment of the superior efficacy of the laser device.
The methodology used in forming the two groups is heterogeneous. It is more useful to select and form groups based on gender and the clinical severity of acne. Considering the heterogeneous etiopathogenic mechanisms dependent on endocrine signaling, the inclusion and exclusion criteria used in this study do not allow for new observations, hypotheses and/or conclusions for clinical practice.
The selected photographs for the two groups are highly heterogeneous. The demographic factors of the patients (race, skin color) are very important.
The type of treatment used in the past is not specified and whether it represented another criterion is not indicated.
The results cannot be stratified based on the dosage and duration of isotretinoin administration.
The images reveal the heterogeneity of the patients, which would explain the need for stratification based on phenotype.
The discussions are not in line with the methodology used in this research. They recall and enumerate certain general data present in the specialized literature.
The conclusions are not supported by the study's results.
Author Response
We sincerely thank the reviewer for their insightful comments.
The Introduction section has been summarised and supplemented with a targeted description of the mechanisms of action of isotretinoin and 675 nm laser, in order to clarify the biological rationale for the combination.
In Materials and Methods, the inclusion and exclusion criteria have been clarified, specifying that patients were divided according to treatment modality, given the exploratory nature of the study.
The Discussion section has been streamlined and reworded to more consistently reflect the methodology adopted and to emphasise the limitations associated with the heterogeneity of the sample.
Finally, the Conclusions have been made more cautious and proportionate, emphasising that the results are preliminary and require further clinical verification.
Reviewer 3 Report
Comments and Suggestions for Authors
The current manuscript entitled "Combining 675 nm Laser with Isotretinoin for Enhanced Acne Vulgaris Treatment Outcomes. " discusses the combination therapy of ablative laser at 675 nm with isotretinoin for the treatment of acne. This is a developing trend in several dermatology conditions but the study needs to be revised in order to be suitable for publication as follows:
- Sample size is too small only 6 which makes the generalization of the results very difficult. authors must state this as a major limitation of the study in the discussion section.
- there is no monotherapy group including isotretinoin only which makes the conclusion that the combination therapy is better than the laser only not accurate. the enhanced effect could be due to the drug only.
- Differences between the two treated groups in terms of the baseline are major which again limit the accountability of the results. the differences in average age 20 # 28, in gender 33 # 89 % females are considered major differences especially in a condition like acne whose severity depends on age and sex to great extent making this differences source of results.
- Isotretinoin dosing must be clearly described in the methodology section
- Minor comments:
- Use the same terminology allover the manuscript: e.g use laser monotherapy instead of laser only therapy sometimes.
- page 2 and 3 require language revision for typos and clarity.
Author Response
We thank the reviewer for his helpful suggestions.
The dosage and duration of isotretinoin therapy are now clearly stated in Materials and Methods
We have standardised the terminology throughout the manuscript by using the term “laser monotherapy”.
The Limitations section explicitly acknowledges the absence of an isotretinoin monotherapy arm and the baseline differences between the groups in age and sex as potential confounding factors.
The language has been revised for clarity and readability
The Conclusions and Limitations now include an explicit reference to the need for prospective studies with isotretinoin only control to clarify the specific contribution and possible therapeutic synergy.
Round 2
Reviewer 1 Report
Comments and Suggestions for Authors
The authors have adequately addressed the previously recommended comments and revised the manuscript accordingly. At this stage, the only remaining issue is the formatting of the table. I recommend adjusting the tables layout to comply with the MDPI template requirements to ensure full consistency with the journal’s formatting guidelines.
Comments on the Quality of English LanguageThe quality of the English language has been sufficiently improved.
Reviewer 2 Report
Comments and Suggestions for Authors
The current form of the article is much improved. The authors carefully followed the suggestions made.
Reviewer 3 Report
Comments and Suggestions for Authors
The authors have addressed all my comments and highlighted the limitations of the study after the discussion section. I think now the study can be accepted as a reporting of their clinical findings to be built on later.